Error estimates for the analysis of differential expression from RNA-seq count data

Burden Conrad J. 1 conrad.burden@anu.edu.au
Qureshi Sumaira E. 1
Wilson Susan R. 1 2
1 Mathematical Sciences Institute, Australian National University , Canberra , Australia
2 School of Mathematics and Statistics, University of New South Wales , Sydney , Australia
Nakai Kenta
Electronic publication date: 2014 Sep 23
Publication date: 2014
Volume: 2
Electronic Location ID: e576
Received 2014 May 22; Accepted 2014 Aug 25
Copyright: © 2014 Burden et al.
Copyright year: 2014
Copyright holder: Burden et al.
License: This is an open access article distributed under the terms of the Creative Commons Attribution License, which permits unrestricted use, distribution, reproduction and adaptation in any medium and for any purpose provided that it is properly attributed. For attribution, the original author(s), title, publication source (PeerJ) and either DOI or URL of the article must be cited.
License URL: https://creativecommons.org/licenses/by/4.0/

Keywords: RNA-seq, Differential expression analysis, False discovery rates

Funding: Australian Research Council Discovery Grants DP1094699 DP120101422 National Health and Medical Research Council NHMRC525453 This project is supported by Australian Research Council Discovery Grants DP1094699 and DP120101422 and National Health and Medical Research Council Grant NHMRC525453. The funders had no role in study design, data collection and analysis, decision to publish, or preparation of the manuscript.

==============================
Background. A number of algorithms exist for analysing RNA-sequencing data to infer profiles of differential gene expression. Problems inherent in building algorithms around statistical models of over dispersed count data are formidable and frequently lead to non-uniform p-value distributions for null-hypothesis data and to inaccurate estimates of false discovery rates (FDRs). This can lead to an inaccurate measure of significance and loss of power to detect differential expression.

Results. We use synthetic and real biological data to assess the ability of several available R packages to accurately estimate FDRs. The packages surveyed are based on statistical models of overdispersed Poisson data and include edgeR, DESeq, DESeq2, PoissonSeq and QuasiSeq. Also tested is an add-on package to edgeR and DESeq which we introduce called Polyfit. Polyfit aims to address the problem of a non-uniform null p-value distribution for two-class datasets by adapting the Storey–Tibshirani procedure.

Conclusions. We find the best performing package in the sense that it achieves a low FDR which is accurately estimated over the full range of p-values, albeit with a very slow run time, is the QLSpline implementation of QuasiSeq. This finding holds provided the number of biological replicates in each condition is at least 4. The next best performing packages are edgeR and DESeq2. When the number of biological replicates is sufficiently high, and within a range accessible to multiplexed experimental designs, the Polyfit extension improves the performance DESeq (for approximately 6 or more replicates per condition), making its performance comparable with that of edgeR and DESeq2 in our tests with synthetic data.

Introduction

High throughput sequencing technologies have largely replaced microarrays as the preferred technology for a number of areas of molecular biology, including gene expression profiling and the detection and quantification of differential gene expression under varying conditions. Transcriptome-wide expression profiling is accomplished via the technique of RNA-sequencing (RNA-seq) in which RNA transcripts sampled from a biological source are fragmented to convenient lengths, reverse transcribed to cDNA, amplified, sequenced and the reads identified by mapping to a reference genome. A concise summary of the RNA-seq procedure is given in the introductory material to Li et al. (2012).

Superficially, RNA-seq data gives the impression of needing little in the way of interpretation: Read counts are a sample of the population of cDNA fragments present and should, in principle, be a direct quantitative measure of the prevalence of the observed sequence in the original biological source. In practice however, there are many sources of both systematic and statistical variability present in the data and further complications related to mapping reads to the reference genome, annotation, and normalisation. Accordingly, a number of software packages have been developed specifically for the purpose of analysing tables of read counts from biological replicate sequencing runs under two or more conditions with the specific purpose of detecting which genes are differentially expressed (DE) and quantifying the degree of differential expression via p-values and estimated false discovery rates (FDRs). An extensive comparison of the performance of eleven such packages has recently been published by Soneson & Delorenzi (2013).

Herein we follow the common convention of using the word ‘gene’ as shorthand for any member of the complete list of expressed sequence tags or transcript isoforms of interest in the transcriptome, and, in common with the packages studied by Soneson and Delorenzi, take as a starting point a table of integer valued read counts after mapping to the transcriptome. When detecting differential expression between two conditions, the list of genes is assumed to partition into a fraction 0 ≤ π0 ≤ 1 satisfying the null hypothesis of no differential expression and a fraction 1−π0 of alternate-hypothesis genes which are DE. The analysis is an application of multiple hypothesis testing and π0 is a priori unknown.

The count data arising from high throughput sequencing technology is well represented as over-dispersed Poisson data, as the Poisson shot noise inherent in sampling a relatively small number of reads from a large number of molecules in solution is compounded with biological variability and with variability due to sample preparation. Over-dispersed Poisson data is typically modelled as power-of-Poisson, implying a mean–variance relationship Var(Yθ) = E(Yθ), or negative binomial (NB), implying a mean–variance relationship Var(Y) = E(Y) + ϕE(Y)2. Choosing an appropriate mean–variance function is critical to achieving accurate FDRs, as this function controls the influence of high-count outliers. Two of the most sophisticated and widely used packages for detecting differential expression from RNA-seq data, namely edgeR (Robinson, McCarthy & Smyth, 2010) and DESeq (Anders & Huber, 2010), model the over-dispersed Poisson count data using a negative binomial model (Robinson & Smyth, 2007; Robinson & Smyth, 2008). The read counts for the biological replicates for each gene in each condition are fitted to a NB distribution via an algorithm that involves borrowing information from count data for the complete set of genes. A transcript abundance for each gene is then inferred from the gene’s NB mean, in combination with a normalisation obtained by matching the distribution of read counts over a subset of genes which are likely not to be DE. The null hypothesis corresponding to no differential expression is that the transcript abundance is the same in both conditions. Both packages provide p-values from which estimates of FDRs are extracted using the Benjamini–Hochberg algorithm (Benjamini & Hochberg, 1995). For concise summaries of differences between the algorithms behind edgeR and DESeq, which are mainly related to the estimation of NB parameters from the raw data see Robles et al. (2012) and the supplementary material to Soneson & Delorenzi (2013). An updated version of DESeq, called DESeq2 (Love, Anders & Huber, 2013) has also appeared, which employs an empirical-Bayes-style shrinkage estimation of over-dispersion similar to that used by edgeR.

Two recent additions to the suite of available packages for analysing RNA-seq data are PoissonSeq (Li et al., 2012) and QuasiSeq (Lund et al., 2012). Both these packages post-date the survey of Soneson & Delorenzi (2013). The PoissonSeq algorithm begins by power-transforming over-dispersed count data to (non-integer valued) quasi-Poisson data. Normalisation is achieved by iteratively determining a subset of genes satisfying a null-hypothesis Poisson model. This subset is typically chosen to be half the total number of genes and is interpreted as falling within the fraction π0 of non-DE genes. An unsigned score statistic, which has a χ2-distribution under the null hypothesis for the Poisson log-linear model described in Section 3 of Li et al. (2012), is used to detect differential expression. The FDR is estimated using a novel modified plug-in estimate in which the permutation distribution of the score statistic is calculated only from genes which are likely to be null. Using evidence of experiments with synthetic NB data, Li et al. (2012) claim that their method achieves considerably improved estimates of the FDR compared with edgeR.

The QuasiSeq package combines a quasi-likelihood approach to estimating over-dispersion (Tjur, 1998) with Smyth’s (2004) approach of sharing information across genes. The QuasiSeq package is implemented as three alternate algorithms, namely QL, QLShrink and QLSpline, which differ in the way in which dispersion information is shared across genes. Of these QLSpline is reported to have the best performance (Lund et al., 2012), and therefore will be the implementation of QuasiSeq used throughout the remainder of this paper.

The current paper introduces an extension to the packages edgeR and DESeq which we call Polyfit. The aim of Polyfit is to improve edgeR and DESeq’s calculations of p-values and estimates of the FDR by replacing the Benjamini–Hochberg procedure with an adapted version of a procedure for multiple hypothesis testing proposed by Storey & Tibshirani (2003).

Our principle purpose is to perform a comparative analysis of the seven packages PoissonSeq, the QLSpline implementation of QuasiSeq, edgeR, DESeq, DESeq2, and our extended versions Polyfit-edgeR and Polyfit-DESeq using both synthetic and real biological data. A secondary aim is to explain details of the Polyfit extension and the issues it addresses. The “Materials and Methods” contains a detailed description of the Polyfit algorithm and our procedure for generating synthetic data. Subsections within the “Results and Discussion” present the results of comparing the performance of the six packages with synthetic and real biological data respectively. A summary of results, including advice on the appropriateness of the packages under differing situations, is given in the “Conclusions”. Throughout the paper we use the default settings, including normalisations, of edgeR, DESeq, DESeq2, PoissonSeq and QLSpline (see Methods for details).

Materials and Methods

Outline of the Polyfit procedure

The software packages edgeR (Robinson, McCarthy & Smyth, 2010) and DESeq (Anders & Huber, 2010) for detecting and quantifying differential expression from RNA-seq data are based on NB models of over-dispersed count data. These packages are state-of-the-art, but nevertheless are subject to shortcomings resulting from the computational complexity of estimating the parameters of the assumed NB distribution for each gene. To illustrate this, Fig. 1A shows histograms of nominal p-values obtained from the DESeq algorithm for simulated data of n = 4 replicates of data for each of two conditions for 46,446 genes created with a range of means and over-dispersions typical of that found in the human transcriptome (Robles et al., 2012). In these data, the mean expression of 15% of the second-condition genes have been up- or down-regulated by at least a factor of 2 relative to the first-condition data. For the purposes of the current illustration, and as part of the implementation of our method, we have made changes to the original DESeq and edgeR algorithms in order to smooth out an artefact spike at p = 1 resulting from estimating p-values from a discrete distribution; further details are given below. We observe that even with this spike redistributed the p-value histogram for the 85% of genes which are unregulated (shaded) is far from uniform. The effect is more pronounced for DESeq than for edgeR. Uniformity is a fundamental property required of p-values for continuous data satisfying the null hypothesis (Storey & Tibshirani, 2003), and hence using the false positive rate to control for differential expression with these calculated p-values would lead to an overly conservative measure of significance and hence loss of power to detect differential expression.

Figure 1 The Polyfit procedure.

(A) Histogram of the nominal p-values calculated by DESeq for synthetic data RNA-seq with 15% genes up- or down-regulated. The shaded histogram superimposed is the 85% of transcripts which are unregulated. (B) Schematic representation of the Storey–Tibshirani procedure for correcting for multiple hypothesis testing, assuming correctly calculated p-values. (C) Schematic representation of the Storey–Tibshirani procedure adapted to RNA-seq data. By ‘nominal p-values’ we mean p-values as calculated by a computer package relying on a NB model using estimated parameters, such as DESeq or edgeR. (TP, true positives; FP, false positives; FN, false negatives; TN, true negatives at a specified significance point α.)

DESeq and edgeR correct for multiple hypothesis testing via the Benjamini–Hochberg procedure (Benjamini & Hochberg, 1995). For each gene an ‘adjusted p-value’ (also known as q-value) is calculated to enable the expected false discovery rate (FDR) (i.e., the proportion of positives returned which are false positives) to be used to control for differential expression. The q-value of an individual hypothesis test is the minimum FDR at which the test may be called significant. Herein we propose an alternate method for estimating p-values and q-values by adapting the graphical procedure for multiple hypothesis testing due to Storey & Tibshirani (2003). In this procedure the proportion of cases satisfying the null hypothesis is estimated from the behaviour of the p-value histogram as p → 1, enabling estimates of q-values to be obtained graphically at any p-value α as the ratio FP/(TP + FP) (see Fig. 1B). The procedure implicitly assumes p-values are calculated accurately and have a uniform distribution under the null hypothesis.

Our proposed adaptation of the Storey–Tibshirani procedure to RNA-seq data method shares features in common with the empirical null method of Efron and its variants (Efron, 2004; Jin & Cai, 2007). The method is illustrated in Fig. 1C and described in detail below. Briefly, a suitable functional form is fitted to the right hand part of the nominal p-value histogram supplied by the existing software and extrapolated to the complete interval [0, 1]. The area under the extrapolated curve is assumed to approximate the histogram of nominal p-values for the non-DE genes. Corrected p-values and q-values are then estimated at each nominal p-value (labelled α in Fig. 1C) from the formulae corrected p-value=FPFP+TN,corrected q-value=FPFP+TP.

The method provides an estimate of the proportion π0 of genes satisfying the null hypothesis of no differential expression as the shaded area divided by the total number of genes, and hence also an estimate of the fraction 1−π0 of DE genes. Herein we refer to our adapted Storey–Tibshirani procedure as ‘Polyfit’ (for polynomial fit).

The ‘Polyfit’ method described in detail below consists of two steps; removing an artefact ‘flagpole’ in the p-value histogram at p = 1, and adapting the Storey–Tibshirani procedure to a non-uniform nominal p-values histogram.

Removal of the ‘flagpole’ at p = 1

Typical p-value histograms produced by DESeq and edgeR in a case where there is no differential expression and a case with 15% differential expression between two conditions A and B with nA = nB = 4 replicates are shown by the red histograms in Fig. 2. These particular examples are for synthetic data generated according to the NB model assumed by DESeq or edgeR. We have observed that qualitatively similar histograms are frequently obtained from real biological data: there is invariably a spike at 1 on top of a distribution which, in the case of no differential expression is rarely close to uniform and is generally skewed towards the right hand end (particularly for DESeq, see also Fig. S20A of Soneson & Delorenzi (2013)). The spike at 1 is an artefact of calculating p-values from a discrete distribution. For our purposes it is convenient to redistribute the spike by approximating the discrete distribution with a continuous distribution.

Figure 2 Nominal p-value histograms generated by DESeq (A and C) and edgeR (B and D) from synthetic data with no DE genes (A and B) and 15% differentially expressed genes (C and D).

Histograms are shown in red before the redistribution of the ‘flagpole’ at p = 1 and in blue after.

For any given gene, p-values are calculated in DESeq and edgeR from a statistic which is a discrete random variable, namely the total number of counts observed in all replicates of condition A conditional on the total number of counts observed in all replicates of both conditions. If the observed total number of counts in conditions A and B for a given gene are kA and kB respectively, and the null hypothesis probability of making this observation is π(kA, kB), then the two-sided p-value is calculated as a sum of probabilities over ways of apportioning the counts which have lower probability than that observed: (1) p=∑a+b=kA+kBπa,b≤πkA,kBπa,b∑a+b=kA+kBπa,b.

If the observed counts happen to hit the mode of the discrete conditional distribution this formula will return a p-value of 1. This is the cause of the spike observed at the right hand end of the red histogram in Fig. 2. This effect is most noticeable for genes with low count numbers. We note in passing that we have also occasionally observed other, much smaller, spikes occurring at rational values in the p-value histogram arising for similar reasons.

By using a method similar to that employed by Marioni et al. (2008), the spikes can be redistributed by replacing the discrete distribution with a ‘squared off’ continuous distribution as follows. Suppose the mass and probability functions of a discrete random variable K∈{0, 1, …, kmax} under a given null hypothesis are (2) Prob K=k=PKk,Prob K≤k=∑a≤kPKa=FKk.

In the current case, kmax = kA + kB and PKk=πk,kmax−k/∑a+b=kA+kBπa,b. For an observed value k of K, a p-value defined as (3) p=2minFKk−1+UPKk,1−FKk−1−UPKk,

where U is a random number with a uniform distribution on the interval [0, 1], will have a uniform distribution under the null hypothesis.

Note that our definition of two-sided p-value has one other difference from that used in DESeq and edgeR, in that Eq. (1) is a sum of probabilities less than or equal to the probability of the observation, whereas Eq. (3) is twice the one-sided p-value. There is no universal agreement on how to define two-sided p-values for asymmetric distributions, and both approaches are common in the literature (see Dunne, Pawitan & Doody (1996) and references therein). Inasmuch as the developers of DESeq and edgeR have sensibly made a convenient ad hoc decision for their purposes, Eq. (3) is an ad hoc decision convenient to the purpose of providing a smooth nominal p-value distribution to enable us to proceed to the adapted Storey–Tibshirani procedure described below.

The blue histograms in Fig. 2 have been produced using DESeq and edgeR software with Eq. (1) replaced by Eq. (3). The spike at 1 and some of the irregularity in the shape of the histogram has been removed, though the underlying skew towards the right remains. The remaining skewing is caused by the need to estimate the parameters of the NB distribution from the data: if the parameters estimated by DESeq for each gene are replaced by the true values used to generate the synthetic data of Fig. 2 for instance, the resulting histogram is very close to uniform (result not shown). In itself, the removal of the spike is mainly cosmetic at this point as the shape of the left hand part of the histogram, which is important for detecting significantly DE genes, remains virtually unaltered. Furthermore, the inclusion of the random number U in the algorithm will cause p-values to vary slightly each time the program is run, particularly for low-count genes. It is probably for these minor reasons that no such technique has been implemented in either DESeq or edgeR. However, implementation is necessary for the following step in the Polyfit procedure.

Corrected p-values and q-values: adapting the Storey–Tibshirani procedure

The principle behind adapting the Storey–Tibshirani procedure (Storey & Tibshirani, 2003) to RNA-seq data is illustrated in Fig. 1C. The major challenge is to estimate the histogram of nominal p-values arising from non-DE genes (the shaded area) given only the histogram of nominal p-values for all genes (the upper curve). This is accomplished by postulating a suitable functional form which is fitted to the right hand part of the histogram over an interval p > λ for an optimised value of λ. The algorithm is implemented in an R function levelPValues() provided in File S2. This function takes as its argument an array of DESeq or edgeR p-values with the spike at 1 redistributed as described above, and generates as output an estimate of the fraction π0 of genes not DE, an array of corrected p-values and an array of corrected q-values. The function also provides a set of plots, an example of which is illustrated in Fig. 3 for synthetic data with 3 replicates in each of two conditions in which 15% of genes are DE in the second condition and p-values are generated from our replacement DESeq function pfNbinomTest().

Figure 3 (A) Estimates of the fraction π0 of genes not DE obtained by fitting a quadratic function to the original input (without the flagpole) p-value histogram over the interval [λ, 1]. (B) Density plot of obtained estimates πˆ0. The optimal λopt (red dot in (A)) is obtained by choosing the πˆ0λ closest to the mode of the πˆ0 density. The mode is also indicated by the dotted line in (A). The original and corrected p-value histograms are shown in (C) and (D), together with optimally fitted quadratic in (C) and its image after correction in (D). The red part of the quadratic is the interval [λopt, 1] in (C) and its image in (D). This example is generated from synthetic data for which the true value of π0 is 0.85.

The algorithm is summarised as follows:

1. For a range of values of λ from 0 to 1 in steps of 0.01 a 3-parameter quadratic function fλ is fitted to the p-value histogram over the interval [λ, 1]. The fit is performed by using the R function nlm() to minimise the sum of squared residuals. An estimate of π0 is obtained at each λ from the formula πˆ0λ=∫01fλxdxtotal number of p-values.

Note that as λ increases past the left-hand peak of the original p-value histogram, which is dominated by DE genes, the fit initially stabilises and is then overcome by noise as λ → 1 (see Fig. 3A).

2. A smoothed density plot dπˆo of the histogram of the πˆ0λ values from step 1 is produced using the kernel density estimator R function density() with default settings (see Fig. 3B). An optimal value λopt is chosen as λopt=arg maxλ dπˆ0λ.

In practice, because πˆ0λ is evaluated at a finite set of λ values, the value corresponding to the πˆ0λ closest to the mode argmax(d) of this density plot is used. The quadratic fλopt fitted over the range [λopt, 1] is shown in Fig. 3C.

3. Corrected p-values are calculated for each of the original p-values p via the formula pcorr=∫0pfλoptxdx∫01fλoptxdx,

and corrected q-values are obtained from the formula qcorr=∫0pfλoptxdxTotal number of p-values <p.

For completeness, the function levelPValues() also provides a set of q-values calculated from pcorr via the Benjamini–Hochberg procedure via the R function p.adjust(). A histogram of corrected p-values is shown in Fig. 3D.

There is nothing in principle to guarantee that the estimate πˆ0 produced by this algorithm will lie in the range [0, 1]. In our investigations with both synthetic and real data we have never observed a case of πˆ0 lying outside the range [0, 1.01], as the (smoothed) p-value histograms produced by DESeq and edgeR invariably resemble the examples in Fig. 2 with, even in the absence of DE genes, a spike a the left hand end. Should a value outside [0, 1] be observed, we recommend running levelPValues() with the option plot=TRUE. The nature of the problem will then be apparent from the plot generated analogous to Fig. 3C.

Note that the flagpole removal step is not appropriate for QuasiSeq, PoissonSeq or DESeq2. Each of these three packages uses a test statistic which is assumed to have continuous distributions under the null hypothesis, so the flagpole problem does not arise: QuasiSeq uses a quasi-likelihood ratio test statistic with a null F-distribution (See Lund et al. (2012), Eq. (2) et seq.); PoissonSeq uses a score statistic which closely follows a null chi-squared distribution (See Li et al. (2012), Eq. (3.9)); and examination of the function nbinomWaldTest() within the DESeq2 software (Love, Anders & Huber, 2013) reveals use of a Wald statistic which has a null normal distribution. Furthermore, the second component of Polyfit, namely the adapted Storey–Tibshirani procedure, is not necessary for QuasiSeq or PoissonSeq as neither suffers from a p-value histogram which rises towards the right hand end. Histograms of p-values in the case of no DE are shown in Figs. S23 and S24 for the various packages. We find in general that if Polyfit is applied to the output of QuasiSeq or PoissonSeq there is no appreciable difference to the resulting p-values or, in the case of synthetic data, to plots of the FDR.

Variants of the Polyfit procedure

We have also tried using cubic and 3- and 4-parameter rational fitting functions, and find that a quadratic to be the most effective fitting function for determining a stable and convincing fit over a range of λ.

As an alternative to the ‘flagpole removal’ step we have also tried constructing the function fλ by fitting over an interval [λ, 0.9] in Step 1 above to avoid the flagpole, and then estimating πˆ0 as above from the fitted quadratic extrapolated over the interval [0, 1]. The results on synthetic data differed in that the estimated FDR was slightly elevated relative to the standard Polyfit procedure (see Fig. S27). We chose not to use this method because of the ad hoc nature of the λ cutoff at 0.9.

Construction of synthetic data

The synthetic datasets detailed in the “Results and Discussion” were created using the method set out by Soneson & Delorenzi (2013) and Robles et al. (2012). Briefly, our synthetic data is based on a NB model of read counts assumed by Robinson & Smyth (2007) and used in edgeR (Robinson, McCarthy & Smyth, 2010) and DESeq Anders & Huber (2010). Each dataset consists of data for two conditions: a ‘control’ set of read counts Kijcontr and a ‘treatment’ set of read counts Kijtreat, for i = 1, …, t genes sequenced from j = 1, …, n replicate cDNA libraries.

For each gene, we begin by providing a pair of NB parameters, the mean μi and over-dispersion ϕi estimated using maximum likelihood from a subset of the Pickrell dataset (Pickrell et al., 2010) of sequenced cDNA libraries generated from mRNA from 69 lymphoblastoid cell lines derived from Nigerian individuals as part of the International HapMap Project (see Robles et al. (2012) for details). The raw reads were mapped onto the human transcriptome (hg18, USCS) using the KANGA aligner (Stephen et al., 2012), and the transcriptome culled to a list of t = 46,446 transcripts (‘genes’ in the above terminology) with an average of at least one count per replicate to reduce the number of zero-count genes. For Figs. S21 and S22 the same maximum-likelihood calculation was followed to estimate NB parameters from the 10 C57BL/6J biological replicates of the Bottomly dataset (Bottomly et al., 2011) consisting of adult mouse-brain RNA-seq reads mapped using the Bowtie aligner (Langmead et al., 2009).

The control read counts Kijcontr were created as independently distributed NB random variables with parameters Λjμi and ϕi. Variability in library size among samples is accounted for by a random scaling factor Λj, which, following Lund et al. (2012), is simulated with a log-normal distribution: log2Λj ∼ N(0, 1). The geometric mean of the total read counts across replicates was ≈∑iμi=1.0×107. Although variability in library size can have a significant effect on method performance (Lund et al., 2012), correlation between genes under simulation schemes like ours does not (Li et al., 2012). To create the treatment data the set of genes is first divided into a non-regulated subset, an up-regulated subset and a down-regulated subset. A regulating factor θi, i = 1, …, t, which is equal to 1 (non-regulated), >1 (up-regulated) or <1 (down-regulated) is then chosen from a suitable distribution. In the current work the regulated genes (1, 5, 10 or 15% of the total) were chosen randomly from the complete set of genes and split into up-regulated and down-regulated subsets of equal size. The regulating factor θi was chosen to be 2 + Xi for the up-regulated genes and (2+Xi)−1 for the down-regulated genes where each Xi is an independent exponential random variable with mean 1. A treatment read count Kijtreat is then generated independently for each isoform in each replicate from a NB distribution with mean θiΛjμi, unchanged dispersion ϕi, and a second independently chosen set of the library scaling factors Λj.

The method construction of synthetic data for Fig. 6 was identical except that for both control and treatment Poisson-inverse-gamma (PIG) data with the appropriate mean and overdispersion was generated in place of NB data.

Software

Our implementation of p-value and q-value calculations using the proposed method in the statistical package R (R Development Core Team, 2013) is provided in File S2, and is currently being developed as a Bioconductor package called Polyfit, which can be downloaded from https://github.com/cjb105/Polyfit. In order to accomplish the ‘flagpole’ redistribution our implementation of the DESeq and edgeR algorithms differs from the original source code in that our function pfNbinomTest() replaces the nbinomTest() in DESeq and pfExactTest() replaces exactTest() in edgeR. The algorithm for calculating p-values and q-values from a redistributed nominal p-value histogram is implemented as the function levelPValues().

In the results described below the original nominal p-values are calculated from DESeq (Anders & Huber, 2010) v1.14.0 with default settings including median of count ratio normalisation, and edgeR (Robinson, McCarthy & Smyth, 2010) v3.4.0 with default settings and using TMM normalisation (Robinson & Oshlack, 2010). The dispersion parameter in edgeR calculations is estimated using the ‘classic’ (as opposed to ‘glm’) routines (Robinson et al., 2013). The version of DESeq2 used is v1.2.5. The version of PoissonSeq (Li et al., 2012) used is v1.1.2. We have observed that this version of PoissonSeq has a potential problem in that the function PS.Main(), which returns p-values and q-values, has the undesirable feature that it resets the seed of the random number generator to the same value in subsequent calls. This may cause problems, for instance, if generating synthetic data and then calling PS.Main() repeatedly within a loop. The version of QuasiSeq (implemented as QLSpline) (Lund et al., 2012) used is v1.0-2. The default normalisations used in the four packages are all based on matching the distribution of read counts over a subset of genes which are likely not to be DE. Such methods have been demonstrated (Dillies et al., 2013) to be superior to normalisations which do not take into account the compositional nature of the data such as Total Count normalisation or Reads Per Kilobase per Million mapped reads (Mortazavi et al., 2008).

Results and Discussion

Synthetic data results

We have tested the performance of seven R packages designed for detecting DE from RNA-seq data using synthetic datasets constructed as described in the “Materials and Methods”. Each synthetic dataset consists of NB distributed counts simulating n replicates of data in each of two conditions in which a specified percentage of genes are DE by at least a factor of 2.

Figure 4 shows estimates of the percentage of DE genes for the cases n = 2 and 4. The PoissonSeq estimate is obtained from the q-value, or estimated FDR corresponding to all genes being called DE. We observe that all methods underestimate the percentage of DE genes when this percentage is sufficiently large and, excepting QLSpline, overestimate the percentage DE when only few genes are differentially expressed. In general QLSpline and Polyfit-edgeR have the best performance, though the Polyfit methods show greater variation between samples, and QLSpline shows very high variation for n = 2 replicates. Note that the original packages DESeq and edgeR on which Polyfit is built do not give direct estimates of the percentage of DE genes.

Figure 4 Estimated percentage 1001−πˆ0 of DE genes for synthetic data representing (A) n = 2 replicates and (B) n = 4 replicates of NB data for two different conditions with a specified percentage of genes differentially expressed by at least a factor of 2 in the second condition.

Polyfit-DESeq and Polyfit-edgeR are labelled with the extension _ PF. Error bars are 90% confidence intervals and points are medians estimated from 100 independently generated datasets. To facilitate simulation of 100 datasets at each data point without undue computational cost, QLSpline was implemented with the option “Model = Poisson”.

Figures 5A, 5C and 5E show true and estimated FDRs (i.e., q-values) calculated from synthetic data over the complete range of p-values and a range of degrees of differential expression for n = 4 replicates of data for two different conditions for each of the seven methods: PoissonSeq (Li et al., 2012), DESeq and DESeq2 (Anders & Huber, 2010), edgeR (Robinson, McCarthy & Smyth, 2010), QLSpline with the option “Model = NegBin” (Lund et al., 2012) and our proposed variants Polyfit-DESeq and Polyfit-edgeR (labelled with the extension _ PF). By ‘true FDR’ we mean the quantity FP/(FP + TP) calculated from the known false postives and true positives out to a total number FP + TP of genes called as being differentially expressed by each package. This will change from package to package because the p-values are ordered differently. The true FDR curves do not differ noticeably on the scale of the plots between DESeq and Polyfit-DESeq or between edgeR and Polyfit-edgeR. By ‘estimated FDR’ we mean the reported “padj” value.

The plots confirm the findings of Li et al. (2012) that PoissonSeq substantially corrects an overestimation of the true FDR by the Benjamini–Hochberg procedure used by edgeR, DESeq and DESeq2 as the significance point is raised to include a large number of genes called as being DE. The plots also show that this shortcoming of packages using Benjamini–Hochberg is rectified by the adapted Storey–Tibshirani procedure which brings Polyfit-edgeR and Polyfit-DESeq into close agreement with PoissonSeq and with the true FDR curves. QLSpline, which uses a method similar to Storey–Tibshirani due to Nettleton et al. (2006), is also in close agreement with the true FDR over the entire range of the plot.

Figure 5 True (solid curves) and estimated (broken curves) FDRs for n = 4 replicates of synthetic negative binomial data in each of two conditions with 5, 10 and 15% genes DE in the second conditon.

Plots (B), (D) and (F) are expanded views of the plots (A), (C) and (E) respectively covering a subset of genes up to a significance point roughly corresponding to the number of DE genes.

Figures S1–S20 show analogous plots for synthetic datasets created with 1, 5, 10 and 15% of genes DE for n = 2, 3, 4, 6 and 10 replicates of data in each of two conditions. For each set of parameter values three independently generated datasets are shown to give an indication of the variation inherent in these simulations. In general we find that provided at least 5% of genes are DE, the Polyfit addition to edgeR and DESeq brings the FDR curves into closer agreement with PoissonSeq and QLSpline and with the true FDR over most of the range of the left-hand plots (A), (C) and (E) of Figs. 5 and 6 and Figs. S6–S20. The agreement between the estimated and true FDRs improves with the number of simulated biological replicates.

An issue not examined in the left-hand plots described above or in the simulations of Li et al. (2012) is the relative performance of different packages and methods for the subset of genes called as being most significant. Each of the right-hand plots (B), (D) and (F) of Fig. 5 and of Figs. S1–S20 is an expanded portion of the neighbouring left hand plot covering the portion of the FDR curves up to a significance point roughly corresponding to the number of DE genes in each simulation: out of a total of ∼46,446 genes, this corresponds to ∼460, ∼2,300, ∼4,600 and ∼7,000 genes for 1, 5, 10 and 15% DE respectively.

Immediately noticeable from these plots are two disadvantages of PoissonSeq, namely that for the genes called as being most significantly DE, the true FDR is consistently higher than for the remaining six methods, and that the true FDR is under-reported by PoissonSeq. This is observed to occur in every case plotted in the right-hand plots (B), (D) and (F) of Fig. 5 and of Figs. S1–S20 for at least half the range plotted. By contrast, for n = 10 replicates, the remaining six methods show an almost zero FDR over half the range plotted irrespective of the percentage of genes DE.

An important point to note is that the choice of NB distribution to construct simulated data may favour packages based on NB models, namely DESeq, DESeq2 and edgeR, over packages not based on NB models, namely PoissonSeq. Accordingly we show in Fig. 6 analogous simulations using Poisson-inverse-Gaussian (PIG) data, which is one of the class of Poisson-Tweedie distributions often used to simulate overdispersed integer-count data. There is evidence from RNA-seq data with very large numbers of replicates that the PIG distribution may be at least as good a fit as NB for many of the genes within the transcriptome (Esnaola et al., 2013). This distribution is closer to Poisson and should therefore not discriminate as much against the package PoissonSeq as the NB distribution. However, Fig. 6 indicates that the relative performance of the packages tested with synthetic PIG data is essentially the same as with NB data.

Figure 6 The same as Fig. 5, except that the synthetic data is generated from a Poisson-inverse-Gaussian distribution.

Considering the right-hand plots (B), (D) and (F) of Figs. 5, 6 and of Figs. S1–S20 in particular, it is clear that the best performing method is QLSpline, for which the estimated FDR closely tracks the true FDR over the whole range of p-values in all cases where the number of replicates is at least 4. Similarly good results are also observed for n = 3 replicates when the percentage of DE genes is >10%. For n = 2 replicates QLSpline overestimates the true FDR, whereas all other methods generally have a tendency to underestimate the true FDR.

The ability of edgeR, DESeq and their Polyfit extensions to estimate the FDR for the genes called as being most significantly DE varies according to the level of differential expression in the synthetic data and the number of simulated biological replicates. Observations from the FDR plots of our simulations out to a significance point corresponding to half the number of truly DE genes are summarised in Fig. 7. At low levels of differential expression (≲ 5% DE) or small numbers of simulated biological replicates (n ≲ 4) all four methods considered in Fig. 7 under-report the true FDR out to the significance point considered in this table. This observation is consistent with the findings reported in Fig. 5B of Soneson & Delorenzi (2013). The Polyfit addition to edgeR and DESeq tends to lower the estimated FDR, thus exacerbating this problem.

Figure 7 Summary of performance of the packages edgeR, DESeq and their Polyfit extensions in estimating the FDR for genes out to a significance point corresponding to half the number of truly DE genes.

(See plots (B), (D) and (F) in Fig. 5 and analogous plots in Figs. S1–S20). Experiments were done with n = 2, 4, 6 and 10 simulated replicates of synthetic data in which 1, 5, 10 and 15% of genes are DE.

For higher levels of differential expression Polyfit proves to be successful for DESeq, but not edgeR. If the level of DE is ≳ 10% and n ≳ 4 simulated biological replicates, DESeq over-reports the FDR over almost the whole range of genes. Polyfit attempts to correct this over-reporting, giving an accurate estimate of the true FDR for n ≳ 6 replicates. However, in the case of edgeR, Polyfit often overcorrects, leading to a more severe underestimate of the true FDR for those genes called significantly DE; see Fig. 7. The problem is less severe for higher numbers of replicates (n ≳ 10), for which there is very little difference between edgeR and Polyfit-edgeR

Polyfit’s over-correction arises because the quadratic fit is unable to capture the small spike at the left-hand end of the histogram of non-DE genes which is visible in shaded portion of Fig. 1A, and which has been observed to occur generally in histograms of non-DE genes for both DESeq and edgeR (Robles et al., 2012). This problem is particularly acute for extremely low levels of differential expression, in which case the quadratic extrapolation means that almost the entire signal of reported differential expression comes from the spurious left-hand spike (see Figs. 2A, 2B), thus limiting the effectiveness of the method in this limit. In general, the performance of DESeq2 is very similar to that of edgeR, which is not surprising given that both packages use similar methods to estimate parameters of the NB distribution for each gene.

In order to check whether our findings are specific to the overdispersion profile of the Pickrel dataset, we have also analysed synthetic datasets generated with mean and overdispersion parameters estimated from the 10 C57BL/6J biological replicates of the Bottomly dataset (Bottomly et al., 2011) described in the Biological data results section below. Figures S21 and S22 show plots of FDRs for n = 4 and 10 simulated biological replicates respectively with 5, 10 and 15% DE genes. We observe behaviour consistent with that of the Pickrell data, namely that in the right-hand plots which show the genes which are called as being most significantly DE, (i) PoissonSeq produces higher number of false discoveries than the other packages, (ii) QLSpline generally provides the most accurate estimate of the FDR, and (iii) the Polyfit extension does not improve edgeR, which has a tendency to under-report the FDR but it does generally improve DESeq which has a tendency to over-report the FDR.

In Table 1 we list the relative CPU time used by each package for the n = 4, 10% synthetic dataset in Fig. 5, using DESeq as a reference. A similar behaviour is observed for the other synthetic datasets analysed. We see that good performance of QLSpline in terms of its ability to estimate FPRs accurately is counterbalanced by a poor performance in terms of CPU time. Note also that the extra overheads imposed by the Polyfit extensions are minor.

Table 1 Relative CPU times for each package to process the n = 4, 10% synthetic dataset in Fig. 5.

Package	CPU time	
PoissonSeq	0.38	
QLSpline	16.45	
edgeR	0.28	
Polyfit-edgeR	0.32	
DESeq	1.00	
Polyfit-DESeq	1.07	
DESeq2	0.82	

In summary, none of the packages considered was able to accurately estimate the true FDR for n = 2 vs. 2 simulated replicates over the range of most significantly called genes up to a portion of the total number of genes equal to the percentage DE. For n > 4 vs. 4 replicates, the overall best performing method (ignoring CPU time) is QLSpline, which for these synthetically generated data accurately estimates the true FDR irrespective of the true percentage of DE genes or the significance point chosen. The next best performing packages are DESeq2, edgeR and the Polyfit extension of DESeq. The Polyfit extension to DESeq will improve the estimate of the true FDR over the complete range of significance points provided the number of replicates and percentage of differentially expressed genes is sufficiently high (≳ 6). We find that edgeR and DESeq2 generally outperform DESeq over the parameter range considered and that the Polyfit extension is effective in correcting an overestimation of the FDR when applied to DESeq, but not edgeR.

Biological data results

Two biological RNA-seq datasets were considered. The first, which we will refer to as the ‘fly data’, originates from experiments by Wilczynski, Liu, Delhomme and Furlong who made their data available to Anders & Huber (2010) ahead of publication for evaluating DESeq, who include it in the supplementary material to their paper. The data consists of n = 2 biological replicates of ‘control’ fly-embryo RNA and 2 replicates of ‘treatment’ RNA in which one gene was engineered to be over-expressed. For our analysis the original dataset of 17,605 genes was culled to remove genes with an average across all replicates in both conditions of one count or less, leaving 13,258 genes. The second dataset, which we refer to as the ‘Bottomly data’, consists of RNA-seq data from 21 adult mouse brains: 10 biological replicates the C57BL/6J strain and 11 of the DBA/2J strain (Bottomly et al., 2011). This dataset is available from http://bowtie-bio.sourceforge.net/recount/. We culled the full set of 36,536 genes to remove genes with an average across all replicates in both conditions of one count or less, leaving 11,123 genes. For both sets of data plots of estimated FDRs against the number of genes called as being DE for each of the seven methods are given in Fig. 8.

Figure 8 Estimated FDRs for (A) the fly data consisting of n = 2 vs. 2 biological replicates of fly-embryo RNA and a total of 13,258 genes, and (B) the Bottomly data consisting of n = 10 vs. 11 biological replicates of mouse RNA and a total of 11,123 genes.

The right hand plots are expanded views of the first few thousand called genes.

Under the assumption that a fraction π0 of genes satisfy the null hypothesis of no differential expression, PoissonSeq, QLSpline, Polyfit-edgeR and Polyfit-DESeq provide estimates of the fraction 1−π0 of DE genes, summarised in Table 2.

We consider first the fly dataset. Consistent with the observations of Lund et al. (2012), the fraction of genes reported as DE varies considerably across methods for this dataset. Figure 4 suggests that for n = 2 replicates the reported fractions are underestimates of the true fraction of DE genes, and that QLSpline’s estimates of π0 are highly variable. With this in mind we compare the fly data FDR curves in Fig. 8A with the results for n = 2 synthetic data with 15% DE genes, Fig. S16. Certain similarities are apparent between the real and synthetic data. In both cases one observes that the edgeR and DESeq curves are higher than their Polyfit counterparts, and that the estimated PoissonSeq and QLSpline FDRs rise sharply compared with the other methods for the first few hundreds called genes. The true FDR curves in Fig. S16 indicate that the FDRs are likely to be overestimated by QLSpline and underestimated by the remaining five methods for the first few hundred most significant genes called DE in the fly data. In particular, the almost zero FDRs reported by DESeq2 and edgeR, DESeq and their Polyfit extensions for the first hundred genes are likely to be overly optimistic estimates.

Table 2 Estimates of the fraction 1−π0 of all genes which are DE for the fly and Bottomly datasets.

	Fly	Bottomly	
PoissonSeq	0.153	0.322	
QLSpline	0.281	0.323	
Polyfit-edgeR	0.144	0.237	
Polyfit-DESeq	0.116	0.104	

Similarly we compare the FDR curves in Fig. 8B for the Bottomly data with the results for n = 10 synthetic data with 15% DE genes, namely Figs. S20 and S22. Once again we observe consistency with the synthetic data results in that the Polyfit addition has reduced the estimate of the FDR, and that PoissonSeq estimates a higher FDR than the other methods for the first few hundred called genes. However, for the real data the Polyfit procedure has not pulled the right hand end of the FDR curves into close agreement, and consequently there is a broad range of estimates of the fraction of DE genes. This discrepancy with the synthetic data may result from a shortcoming of the idealised synthetic data construction which divides the genes into an absolutely non-DE fraction π0 and a remaining fraction 1−π0 which are DE by at least a factor of 2. In reality all genes may be DE to some extent, with a degree of differentiation ranging continuously from near-zero for the majority of genes to considerably non-zero for a small minority of genes. Under these conditions the histogram of p-values does not necessarily split unambiguously into DE and non-DE components. Note also, that for the first 200 called genes, all methods except PoissonSeq report a FDR of almost zero. The evidence in Figs. S20 and S22 from the synthetic data is that this may indeed be an accurate representation of the true FDR.

Figures S25 and S26 show Venn diagrams of the genes which are called DE by QLSpline, Polyfit-edgeR and Polyfit-DESeq up to cutoffs of 100 and 500 most significantly called genes for each of the two datasets. In each case the degree of overlap is reasonably good, with approximately 75% of genes called by any one package common to all three methods.

Conclusions

We have surveyed the effectiveness of a number of software packages designed for two-class detection of differential expression from RNA-seq data via the use of synthetically generated datasets similarly to Soneson & Delorenzi (2013). The packages, edgeR (Robinson, McCarthy & Smyth, 2010), DESeq (Anders & Huber, 2010), DESeq2 (Love, Anders & Huber, 2013), PoissonSeq (Li et al., 2012) and the QLSpline implementation of QuasiSeq (Lund et al., 2012) are all based on statistical models of over-dispersed Poisson data, which, in the case of edgeR and DESeq is expicitly modelled as NB data. Our survey uses synthetic NB data with a range of parameters estimated from the Pickrell dataset (Pickrell et al., 2010) to assess the FDR achieved and the ability of each method to accurately estimate the FDR.

To avoid clutter on the graphs we did not include the other methods given in Soneson & Delorenzi (2013), namely NBPSeq, TSPM, baySeq, EBSeq, NOISeq, SAMSeq, ShrinkSeq, voom (+ limma) and vst(+ limma) as the comparisons of these approaches with edgeR and DESeq are adequately covered there. We note that Soneson & Delorenzi (2013) report that good FDR control was achieved by vst(+ limma). This is a hybrid method constructed by combining the variance-stabilising transformation provided by DESeq with a linear fit to the resulting nomalised counts using limma. Unfortunately the code given in the Supplementary material to Soneson & Delorenzi (2013) to perform these combined tasks does not function if DESeq2 is loaded as the R function getVarianceStabilizedData() has been overwritten with a more recent version which is incompatible with DESeq. Furthermore, we have found in test simulations that DESeq2 has a similar performance to vst(+ limma), but without the complication of being a hybrid of distinct packages.

We have also introduced an add-on to the NB-based packages edgeR and DESeq for two-class detection of differential expression called Polyfit which achieves two of the advantages associated with the recently introduced packages PoissonSeq and QuasiSeq. Firstly, assuming that the transcriptome partitions unambiguously into a fraction π0 of non-DE genes and a fraction 1−π0 of DE genes, Polyfit gives an estimate of π0 (that is an empirical extension of the Storey–Tibshirani algorithm (Storey & Tibshirani, 2003)) which performs at least as well as PoissonSeq and comparably with QLSpline in experiments with synthetic data (see Fig. 4). Secondly, by adapting the Storey–Tibshirani algorithm Polyfit gives a more accurate estimate of the FDR than the Benjamini–Hochberg procedure used by edgeR and DESeq over the central and right-hand sections of the p-value spectrum (see the left-hand plots (A), (C) and (E) in Figs. 5 and S1–S20).

Of more immediate interest to practising biologists is the software’s performance for the genes called as being most significant, that is, the few hundred or so genes with the lowest p-values (see the right-hand plots (B), (D) and (F) in Figs. 5 and S1–S20, which are an expanded view of the left hand plots). Our experiments with synthetic NB data indicate that for these genes the best performing method of those tested is QLSpline, both in the sense that the achieved FDR is among the lowest, and that provided at least 4 replicates are used in each condition the q-values quoted accurately reflect the true FDR over the whole range of p-values. The performance of QLSpline is less consistent for n = 3, but still better than the other methods. The worst performing method is PoissonSeq, which, for the most significantly called genes, consistently achieves high but under-reported FDRs. As a general rule we would not recommend experimental designs with n ≤ 2 replicates in each condition as none of the methods tested is able to estimate the FDR consistently or accurately.

The performance of edgeR and DESeq lies somewhere between these extremes of QLSpline and PoissonSeq. The performance of DESeq2 is very similar to edgeR, probably due to the fact that both packages use similar methods to estimate parameters of the model NB distribution. For the most significantly called genes, the true FDR of edgeR and DESeq is comparable with that of QLSpline, and very low compared with PoissonSeq. Because the Polyfit extension shifts the reported p-value but makes very little difference to the order of the p-values, except for a few genes with very low counts, it leaves the true FDR virtually unchanged.

The main intention of developing Polyfit was to improve the calculation of p-values and q-values (the estimate of the FDR) reported by DESeq and edgeR. As explained in the “Materials and Methods”, Polyfit achieves a close-to-uniform distribution of p-values for the non-DE genes. Polyfit’s ability to estimate the FDR for the genes called as being most significantly DE is summarised in Fig. 7. In their original forms, for low numbers of simulated biological replicates in each condition edgeR and DESeq underestimate the true FDR for the genes called as being most significantly DE. The Polyfit extension tends to lower the estimated FDR thus exacerbating this problem. On the other hand, for higher numbers of simulated biological replicates DESeq overestimates the true FDR and the Polyfit extension can give a more accurate estimate of the FDR if sufficiently many biological replicates are used. Our numerical simulations indicate that with 15% of genes truly DE, Polyfit will give an improved and acceptably accurate estimate of the true FDR for DESeq with n ≳ 6 replicates in each condition. Although this exceeds the number of replicates used in many current RNA-seq experiments, we note that the cost of sequencing is continually decreasing. Furthermore, simulations with synthetic data by Robles et al. (2012) demonstrate that sacrificing sequencing depth to process more replicates by multiplexing leads to considerable gains in the power to detect DE. Unfortunately, for the reasons detailed in the “Synthetic Data Results” the Polyfit procedure fails to improve the edgeR estimate of the FDR over the range of genes called as being most significant.

We have also applied all six methods to two real biological datasets, the ‘fly data’ reported in Anders & Huber (2010) with n = 2 replicates in each condition and the ‘Bottomly’ mouse-brain data (Bottomly et al., 2011) with n = 10 and 11 replicates respectively in the two conditions. The relative differences in reported FDRs between the various methods is observed to follow qualitatively similar behaviour for real biological data as for the synthetic data (see Fig. 8), and the overlap between the first few hundred genes called as being significant between QLSpline, Polyfit edgeR Polyfit DESeq is approximately 75% (see Figs. S21 and S22).

Based on the above simulations, if CPU time is not an issue (see Table 1), we would in the first instance recommend the QLSpline implementation of QuasiSeq with an experimental design having no less than n = 4 replicates in each condition. However, in the interests of confirming experimental analysis with more than one package, and given the cost benefits of multiplexing, we would further recommend also using edgeR, DESeq2, or the Polyfit extension to DESeq with at least 6 biological replicates in each condition. Abbreviations

DE differentially expressed

FDR false discovery rate

NB negative binomial

PIG Poisson-inverse-Gaussian

RNA-seq RNA-sequencing

Supplemental Information

File S1 Supplementary figures S1–S27

Click here for additional data file.

File S2 R code for the Polyfit software, together with, as an example, the code used for generating Fig. 5, together with a data file of mean and overdispersons used for generating the various synthetic data sets used in this paper

Click here for additional data file.

Additional Information and Declarations

Competing Interests

Author Contributions

The authors declare there are no competing interests.

Conrad J. Burden conceived and designed the experiments, analyzed the data, contributed analysis tools, wrote the paper, prepared figures and tables, reviewed drafts of the paper.

Sumaira E. Qureshi analyzed the data, prepared figures and tables, reviewed drafts of the paper.

Susan R. Wilson conceived and designed the experiments, reviewed drafts of the paper.

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
