# Peer review of "Error estimates for the analysis of differential expression from RNA-seq count data"

_PeerJ, doi:10.7717/peerj.576_

## Round 0.1 · original submission · Major Revisions

As you will find from the reviewers' comments, all of them are basically interested in the topic of the manuscript but also raise substantial criticism, some of which overlap each other. Please read the following reviews carefully and revise the manuscript accordingly.

Reviewer 1 ·

Basic reporting

No Comments

Experimental design

No Comments

Validity of the findings

No Comments

Additional comments

The authors compared several computer programs, which were proposed for the analysis of finding differentially expressed genes and proposed Polyfit to achieve flat histgram of p-values that could yield more precise estimation of q-values. Under the over-dispersion case, it could be occurred that the histgram of p-values does not uniform when the null hypothesis is true. So, more false positives will be generated and the estimated q-values could be more conservative. I evaluated this paper and it would have two advantages: One is that the comparision of differential analysis of RNA-seq data is important. The other is that precise estimatin of q-value is important for large data analysis. Due to these two reasons, I think this paper has the potential to be published from PeerJ, but I suggest the authors consider the following points for revision:
- The authors considered over-dispersion is the reason of non-uniform histgram. But I think dependency of hypotheses can yield non-uniformity and RNA expressions of genes are unsually and strongly correlated with each other.
- The authors considered sufficient number of replicates. This is very important in practical situations. But I think it depends on the depth of RNA-seq data. I could not find the information of depth in synthetic data.
- The results may depend on aligner. If you use tophat, do the results change?
- The authors fit quadratic funtion to remove peak at p=1. Why quadratic? This idea may come from ST algorithm for FDR, but they fit splines. This is just a comment, but do you have some theoretical or empirical justification to the use of quadratic function?
- In page 6, \lambda_{opt} is not defined mathematically.
- In real example, would you summarize how your Polyfit improves the analysis results, in terms of biological discussion to the detected genes, quantatively.

Reviewer 2 ·

Basic reporting

In this manuscript the authors report their findings on an evaluation of several existing RNA-Seq analysis methodologies in addition to their modification to two of the methodologies, DESeq and edgeR. The modification introduced by the authors is an attempt to correct for the inaccurate estimation of FDR, due to the estimation of p-values from a discrete distribution.

In general the paper is well-written and easy to follow.

Experimental design

In describing how they used the Storey-Tibshirani procedure to correct the p-values, the authors mentioned the fitting of a quadratic function to the p-value histogram. However, the author did not include an discussion of the method used to fit the quadratic function, and also did not analyze the cost of the additional overhead added to DESeq and edgeR when using this method to correct the p-values.

For the analysis of the synthetic data, there should also be a discussion of the comparison of the different methods for datasets generated using different over-dispersion profile. Specifically, the authors used the estimates for the means and over-dispersion parameters estimated for each gene from the Pickrell dataset. However, it is not clear if their findings would hold if the parameters were estimated from a different dataset having different parameter profiles for the genes.

Validity of the findings

In Figure 4, it would help the discussion for the benefits of using the Polyfit modification for DESeq and edgeR if the plots for the original DESeq and edgeR results are also presented.

Plots in Figure 6 is quite cluttered with lines, and with the color choices it's rather difficult to compare the curves for the different methods.

Overall the paper presents an interesting comparison of the different RNA-Seq DE analysis methodologies. However, regarding the Polyfit modification, current discussion lacks mentioning of the overhead added to the original algorithms. Given that from the manuscript the original algorithms do not benefit much from Polyfit until higher number of replicates, it is important to add an discussion about whether the small benefits are worth the additional computational power required.

·

Basic reporting

The authors proposed a method to adjust FDR and assessed several methods for differential analysis, and demonstrated QLSpline implementation of QuasiSeq produces the closest FDRs to the true ones on synthetic data within a limited number of methods. The idea of FDR correction itself is interesting. However, I have several concerns in this manuscript (i) it is not clear if the authors aims to assess the individual methods or propose their method, (ii) it is difficult to see the authors' claim is supported by their results.

# Major

Regarding the point (i):
* Figure 4 dose not provide any comparison of with and without Polyfit function. The manuscript does not touch application of Polyfit to QuasiSeq or PoissonSeq. If a goal of this manuscript is a demonstration of Polyfit, these comparison has to be performed.
* The mansucript don't include any methods of valiance stabilizing transformation, which is highlighted by Soneson and Delorenzi as good performance for a reasonable number of replicates, while the paper is extensively cited. If a goal of this manuscript is comparison of differential analysis methods, I don't understand why any of these methods are not evaluated.

Regarding the point (ii):
* The authors claims "the Polyfit addition to edgeR and DESeq brings the FDR curves into closer agreement with PoissonSeq and QLSpline and with the true FDR over most of the range of number of genes called as being DE. The agreement between the estimated and true FDRs improves with the number of simulated biological replicates (9/19, bottom 3 lines)". However, I don't find any evidence in Figure 5, 15% DE. For example, the addition of Polyfit to edgeR shifted the result further aggressive, in contrast to the 'true' FDR.
* The authors claims "The Polyfit procedure attempts to correct this over-reporting, the effect of which is to give an accurate estimate of the true FDR for sufficiently high numbers of biological replicates (n >= 10 for edgeR or n >= 6 for DESeq) (12/19, 6th paragraph)". However I see the edgeR and edgeR_PF is almost the same in Figure S20.
* The figure 7 claims edgeR or edgeR_PF can be accurate in a few settings of parameters. However, I don't see such accuracy (consistency with edgeR_True) even in Figure S15 and S20.

# Minor:

* In the 4/19, the authors explained the reason of spike at 1, which is likely methods' artefacts. The manuscript describes the formula when it produce p value = 1, but it does not explain why p value = 1 happens so frequent.

Experimental design

The goal of the experimental design is not clear as commented in "Basic Reporting" (i)

Validity of the findings

The findings are not clearly demonstrated as commented in "Basic Reporting" (ii)

---

## Round 0.2 · Minor Revisions

Your revised manuscript has been reviewed by the three original reviewers. As you will find, two of them are satisfied with the revision while one reviewer still raises a minor point. Please read the comment carefully and let me know how you will respond.

Reviewer 1 ·

Basic reporting

I'm satisfied with the revision.

Experimental design

I'm satisfied with the revision.

Validity of the findings

I'm satisfied with the revision.

Additional comments

I'm satisfied with the revision.

Reviewer 2 ·

Basic reporting

No Comments

Experimental design

No Comments

Validity of the findings

No Comments

Additional comments

I believe the authors have adequately addressed the questions that I raised in the original review. I would recommend that the paper be accepted at PeerJ.

·

Basic reporting

Most of my comments have been addressed properly. Only one remaining point is description about "flagpole". The authors explained that "P value = 1" happens when genes hit the mode of discrete distribution, and "flagpole" (frequent events of P value = 1) is caused by a large number of lowly expressed genes (hit the mode with hight probability). This does not explain why "flagpole" happens only in edgeR and DESeq2, not in PoissonSeq and QuasiSeq, while the latter two methods are based on discrete distribution too. I suggest the authors to mention this point additionally, and add a (supplemental) figure about P-value distributions (such as Figure 2) for PoissonSeq and QuasiSeq demonstrating no "flagpole" for them.

Experimental design

Well designed.

Validity of the findings

Supported by the data.

---

## Round 0.3 · accepted · Accept

I confirm that you have addressed the last raise point properly and thus am happy to accept the manuscript. Congratulations and thank you for your support to PeerJ!